# Activation of allylic esters in an intramolecular vinylogous kinetic resolution reaction with synergistic magnesium catalysts

Dan Li[1], Yuling Yang[1], Minmin Zhang[1], Linqing Wang[1], Yingfan Xu[1], Dongxu Yang [1✉] & Rui Wang [1✉]

Kinetic resolution (KR) of racemic starting materials is a powerful and practical alternative to prepare valuable enantiomerically enriched compounds. A magnesium-catalyzed kinetic resolution based on a designed intramolecular vinylogous Michael reaction is disclosed. Here we show a synergistic catalytic strategy based on the development of chiral ligands. Substrates containing linear allylic ester structures are designed and synthesized to construct key [6.6.5]-tricyclic chiral skeletons via this kinetic resolution process. Detailed mechanistic studies reveal a rational mechanism for the current intramolecular vinylogous KR reaction. The desired direct intramolecular asymmetric vinylogous Michael reaction of linear allylic esters is realized in high efficiency and enantioselectivity with the synergistic catalytic system.

---

[1] Key Laboratory of Preclinical Study for New Drugs of Gansu Province, Institute of Drug Design & Synthesis, School of Basic Medical Sciences, Lanzhou University, Lanzhou 730000, China. ✉email: yangdx@lzu.edu.cn; wangrui@lzu.edu.cn

Catalytic nonenzymatic kinetic resolution (KR) of racemic starting materials that mediates the selective reaction of one enantiomer has been recognized as a powerful and practical alternative to preparing valuable enantiomerically enriched compounds, and found wide applications in both academia and industry[1–4]. Most documented nonenzymatic KR reactions use intermolecular pathways, with the selection of one appropriate reactive partner to finish the desired KR process. In contrast, the development of intramolecular KR reactions have been relatively less investigated, as they require an ideal match between the designed substrates and small molecule catalysts[5–14]. Herein, we design an intramolecular vinylogous Michael reaction of linear allylic esters for a KR process to build chiral parallel [6.6.5]-tricyclic skeletons, which exist in many natural products and pharmaceutically active compounds, such as Juglocombin B, Glaziovianol, and some COX-2 and ubiquitin-connected enzymes inhibitors (Fig. 1)[15–19]. This reaction also represents one alternative to asymmetric dearomatizative pathways of 1-naphthols[20–26].

Compared with other linear allylic carbonyl compounds, simple linear allylic esters are less reactive and less investigated in asymmetric reactions[27–29]. To date, there are still very few studies on the direct activation of linear allylic esters in catalytic asymmetric reactions. Moreover, the α-position of linear allylic esters might dominate the C-C bond formation process especially in the reaction with Michael acceptors[30–32]. In most cases, activated or modified allylic esters are often necessary to overcome the low reactivity of these types of substrates[33–39]. For example, in the widely used vinylogous Mukaiyama reaction, it is necessary to prepare the unstable dienolsilanes in a separate step[33–37]. Only until very recently, the Yin group reported direct asymmetric vinylogous aldol reactions of allylic esters using chiral copper catalysts and additive bases[40]. They also achieved the asymmetric

alkynylogous aldol reaction by an optimized propargyl copper(I) catalytic method[41]. These reactions are highly efficient and ideal for the direct use of allylic esters as feedstock. However, the direct catalytic asymmetric intramolecular vinylogous reaction of allylic esters has not yet been achieved[42–44]. Herein, by developing a synergistic in situ generated magnesium catalytic strategy[45–55], we successfully employ the vinylogous Michael reaction of linear allylic esters in a rationally designed intramolecular KR process (Fig. 1).

## Results

**Reaction optimization.** Initially, we designed and synthesized the allylic ester **1a** for the intramolecular KR reaction. Bifunctional diols containing amine groups (Fig. 2) were selected as chiral ligands for sequencing process to the magnesium catalysts. The desired intramolecular vinylogous Michael reaction proceeded primarily from one enantiomer, and resulted in enantiomerically enriched parallel [6.6.5]-tricyclic skeletons (Table 1). Different tertiary amines-modified diol ligands were screened, and pyrrolidine-modified ligand **L1** had better resolution results compared with those of other tertiary amine groups (Table 1, entries 1-5). Further modification at the 6,6'-position of the BINOL skeletons led to the successful synthesis of a series of bifunctional chiral ligands (Fig. 2, **L7-L10**). These modifications dramatically affected the efficiency of the magnesium catalysts, and the introduction of chloride was identified as giving the best results for the intramolecular vinylogous KR reaction (Table 1, entry 10). The synthetic route for ligand **L10** is illustrated in Fig. 2.

**Substrate scope.** Next, we investigated the scope for the intramolecular vinylogous KR reaction (Fig. 3). The magnesium

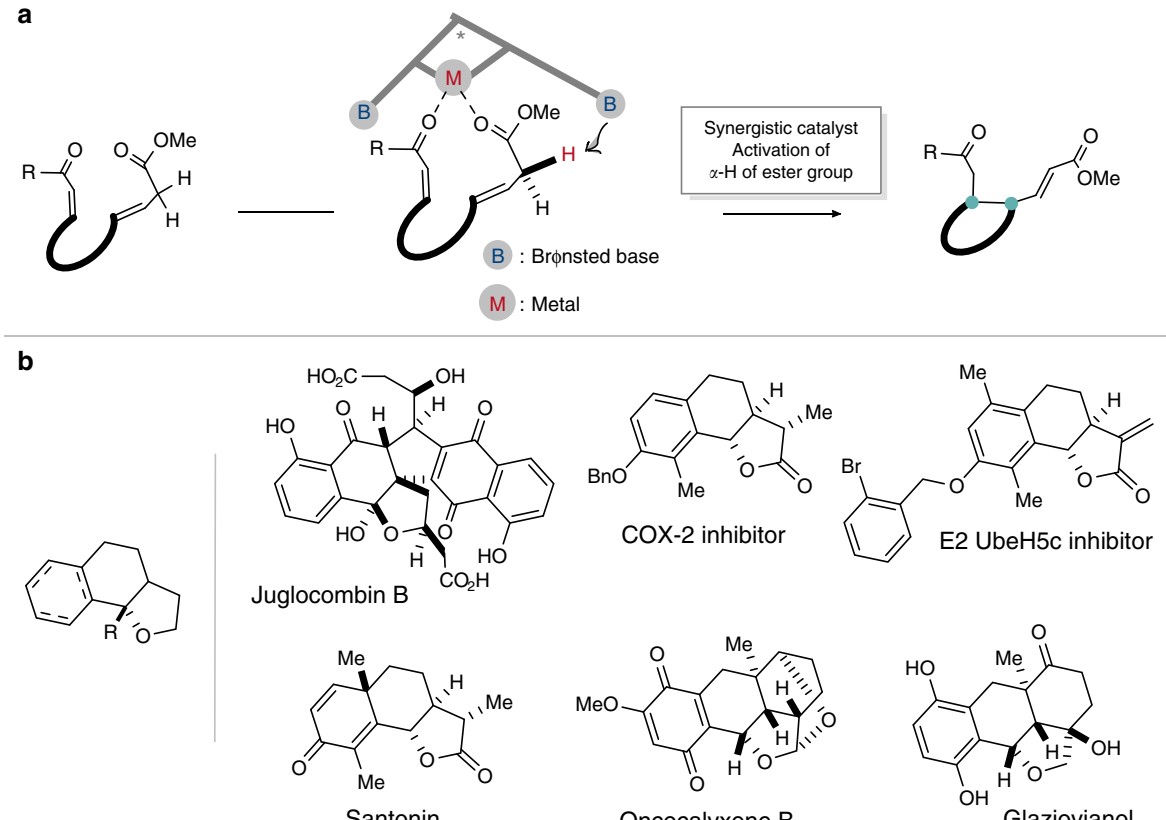

**Fig. 1 Reaction design and related compounds containing the key [6.6.5]-tricyclic skeletons. a** Synergistic catalytic strategy for the direct intramolecular asymmetric vinylogous Michael reaction. **b** Related natural products and pharmaceutically active compounds containing the tricyclic skeletons.

**Fig. 2 Selection and development of bifunctional chiral ligands and related synthetic methods. a** Bifunctional chiral ligands screened in the optimization process. **b** Synthetic method of chiral ligand **L10** and related X-ray analysis.

catalytic system proved to be robust for the selective conversion of different alkyl-substituted substrates, furnished the desired KR process smoothly under mild conditions. A variety of substituted [6.6.5]-tricyclic rings were obtained in high enantioselectivities (92:8-98.5:1.5 er), and the enantiomerically enriched allylic esters **1\*** were recovered in satisfactory results. Substrates with aryl groups also finished the designed KR process, although relatively higher catalyst loading (20 mol%) was required (Fig. 3). The absolute configuration of the resolution adducts was determined by the X-ray crystallographic analysis of **2b** (Fig. 3).

Subsequently, different benzohexene ketone motifs were introduced into the allylic ester substrates and used in the vinylogous KR reaction. Polycyclic structures were established under the magnesium catalytic system. Electron-withdrawing or electron-donating groups were under trial in the KR process (Fig. 4).

Interestingly, it was observed that for substrate **1t**, bearing two Michael receptor sites, the vinylogous Michael reaction occurred during the KR process to form the quaternary stereocenter, and generate the bridged-ring adduct **2t**[56]. In addition, some of **1t\*** was recovered at a moderate er value. The common cyclization adduct **2t'** was not observed under the catalytic system, instead, some undetermined decomposition products were generated, resulting in the relatively lower yields of **2t** and **1t\*** (Fig. 5).

**Transformations**. The vinylogous KR reaction was then carried out at the gram scale and transformations of the recovered **1a\*** were conducted. As illustrated in Fig. 6, the recovered substrate **1a\*** formed the cyclization adduct **2a'**, by treatment with NaOMe.

Under photocatalytic conditions lead to the polycyclic product **3** after finishing the [2 + 2] cyclization process (Fig. 6)[57–60].

To our pleasure, the rearomatization reaction was easily realized by treating **2a** with p-toluenesulfonic acid under mild conditions, This reaction can be used for the formal construction of γ-arylation adduct **4** with high enantioselectivity and good yield (Fig. 7).

Additional transformations of the tricyclic rings were performed for this central [6.6.5] skeleton to form compounds that might be useful for pharmaceutical investigations[15–19]. We introduced different functional groups or heterocyclic structures to the central skeletons by selected cross-coupling reactions. These transformations were carried out by established transition-metal mediated coupling reactions as illustrated in Fig. 7.

**Mechanistic studies**. To investigate mechanistic aspects of the intramolecular vinylogous KR reaction, we performed a variety of mechanistic experiments. We first performed control experiments to identify the reason for the high efficiency of the bifunctional magnesium catalyst. As illustrated in Fig. 8, simple in situ generated magnesium catalyst from BINOL cannot promote the intramolecular vinylogous reaction (Fig. 8, a). Introduction of tertiary amine at high loading mediated generation of the trace cyclization adduct, and the combined use of the BINOL-Mg catalyst and tertiary amine activated the allylic ester **1a** to form intramolecular vinylogous Michael adduct **2a**. These results indicate the magnesium center and the Brønsted base can synergistically activate the designed allylic ester substrate. The developed bifunctional magnesium catalyst is more effective in the vinylogous KR reaction even with the ligand **L12** with lower

**Table 1 Optimization of the vinylogous KR reaction[a].**

| Entry | L | Solvents | 1a* (er)[b] | 2a (er)[b] | s[c] |
|---|---|---|---|---|---|
| 1 | L1 | CPME | 47 (78.5:21.5) | 30 (94.5:5.5) | 31 |
| 2 | L2 | CPME | 82 (55:45) | 15 (80:20) | 4 |
| 3 | L3 | CPME | 82 (55:45) | 15 (80:20) | 25 |
| 4 | L4 | CPME | 41 (87:13) | 31 (92:8) | 5 |
| 5 | L5 | CPME | 72 (57.5:42.5) | 11 (88:12) | 8 |
| 6 | L6 | CPME | 61 (51:8) | 14 (55:45) | 1 |
| 7 | L7 | CPME | 50 (52.5:47.5) | 40 (95:5) | 33 |
| 8 | L8 | CPME | 49 (58.5:21.5) | 43 (83.5:16.5) | 9 |
| 9 | L9 | CPME | 59 (74.5:25.5) | 32 (85:15) | 9 |
| 10 | L10 | CPME | 45 (94:6) | 46 (98.5:1.5) | 192 |
| 11 | L10 | toluene | 45 (85.5:14.5) | 38 (95.5:4.5) | 45 |
| 12 | L10 | o-xylene | 38 (87:13) | 35 (96:4) | 53 |
| 13 | L10 | CHCl$_3$ | 73 (54:46) | 9 (90.5:9.5) | 10 |
| 14 | L10 | THF | 76 (51:49) | 10 (92.5:7.5) | 12 |

CPME, cyclopentyl methyl ether; THF, tetrahydrofuran.
[a]Reactions were carried out with **1a** (0.2 mmol) with Bu$_2$Mg and ligands (10 mol%), in solvents (0.2 M) for 9 h.
[b]Isolated yields of **1a*** and **2a** were reported and er values were analyzed by chiral HPLC.
[c]s = ln[(1 − C)(1 − ee)]/ln[(1 − C)(1 + ee)], where ee = ee$_{1a*}$/100, ee$_{1a*}$ = (R$_{1a*}$ − S$_{1a*}$)/(R$_{1a*}$ + S$_{1a*}$)*100, C is monitored by HPLC analysis and calculated according to C = ee$_{1a*}$/(ee$_{1a*}$ + ee$_{2a}$).

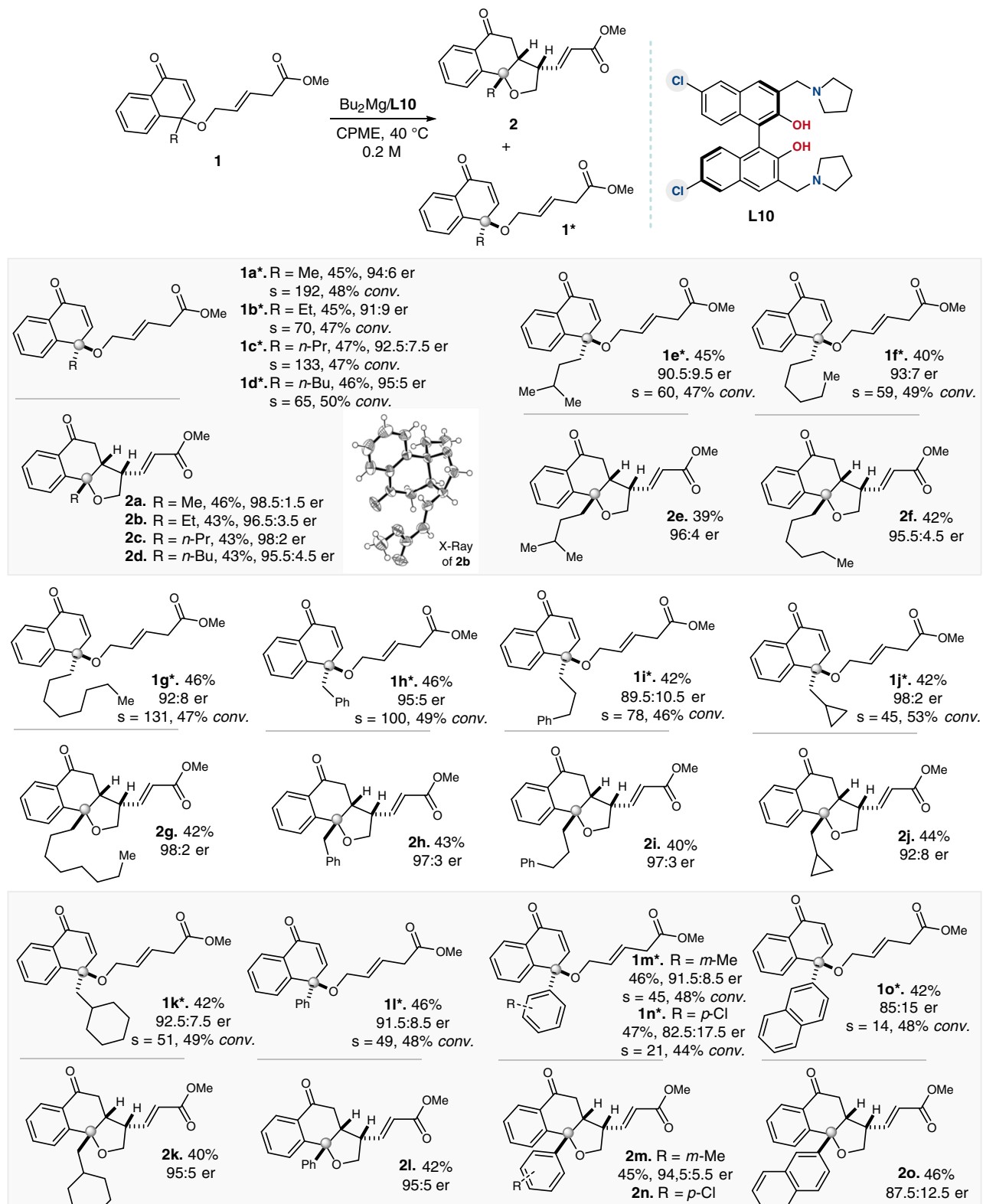

**Fig. 3 Substrate scope of the KR reaction.** See Supplementary Information for the detail experiment processes. All yields shown were based on isolated products. er values were determined by chiral HPLC analysis. $s = \ln[(1 - C)(1 - ee)]/\ln[(1 - C)(1 + ee)]$, where $ee = ee_{1a*}/100$, $ee_{1a*} = (R_{1a*} - S_{1a*})/(R_{1a*} + S_{1a*})*100$, C is monitored by HPLC analysis and calculated according to $C = ee_{1a*}/(ee_{1a*} + ee_{2a})$.

**Fig. 4 Further extensions of the substrate scope of the KR reaction.** All yields shown were based on isolated products. er values were determined by chiral HPLC analysis.

**Fig. 5 Site-selective results of the substrate 1t in the KR reaction.** Isolated yields are reported.

Brønsted basicity (Fig. 8, a). Subsequently, studies of nonlinear effects revealed the synergistic catalyst interacts with the bidentate substrate as a mono-species (Fig. 8, b)[61,62]. Further investigations on ESI experiments of the initial reaction complexes clearly indicated the coordination results of the immediately introduced bidentate substrate 1a to the bifunctional magnesium catalyst, which is well in accordance with the calculated results (Fig. 8, c).

**Proposed mechanism.** Combination with the mechanistic insights, a possible mechanism cycle of the intramolecular vinylogous KR reaction is proposed (Fig. 9). The bifunctional magnesium catalyst is smoothly generated from L10 and Bu$_2$Mg after the neutralization process, then the bidentate substrate coordinates to the magnesium center and the tertiary amine synergistically promotes enolation of the allylic ester. At the same time, the bidentate coordination results in synchronous activation of the Michael receptor to promote the intramolecular vinylogous reaction in the well-controlled chiral environment (Fig. 9, II and III). Finally, the protonation process and the entry of another molecule of 1a lead to the release of the KR product 2a.

## Discussion
In summary, we have accomplished a direct catalytic asymmetric intramolecular vinylogous Michael reaction. Bifunctional chiral

**Fig. 6 Gram scale trial and related transformation of 1a\*. a** Gram scale experiments of the KR process. **b** Transformations of the isolated chiral allylic ester **1a\***.

**Fig. 7 Transformations of resolution products.** Conditions: **a** with CuI (2.5 mol%), Pd(PPh₃)₄ (2.5 mol%), Et₃N (2.0 equiv) in DMF at room temperature. **b** with PPh₃ (20 mol%), Pd(OAc)₂ (10 mol%), Et₃N (2.0 equiv) in THF/CH₃CN at room temperature. **c** with PPh₃ (6 mol%), PdCl₂(PPh₃)₂ (3 mol%), K₂CO₃ (1.5 equiv) in Dioxane at 80 °C. **d** with Pd(PPh₃)₄ (5 mol%), K₂CO₃ (2.0 equiv) in H₂O/ Dioxane reflux for 24 h.

ligands were developed to generate synergistic magnesium catalysts. Using the designed allylic ester substrates, the KR process successfully led to the expected [6.6.5]-tricyclic key skeletons. Several transformations were conducted to give types of chiral polycyclic structures and derivatives of the [6.6.5]-tricyclic skeletons, as well as the enantioselective γ-arylation adduct. Combinational mechanistic insights, including control experiments, nonlinear effects studies and relative ESI investigations, led to the

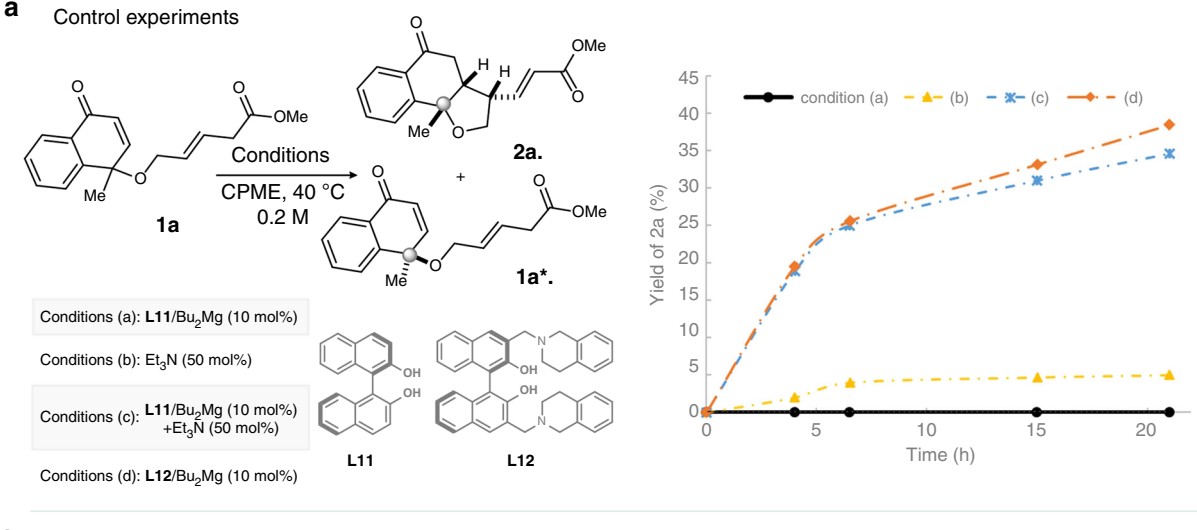

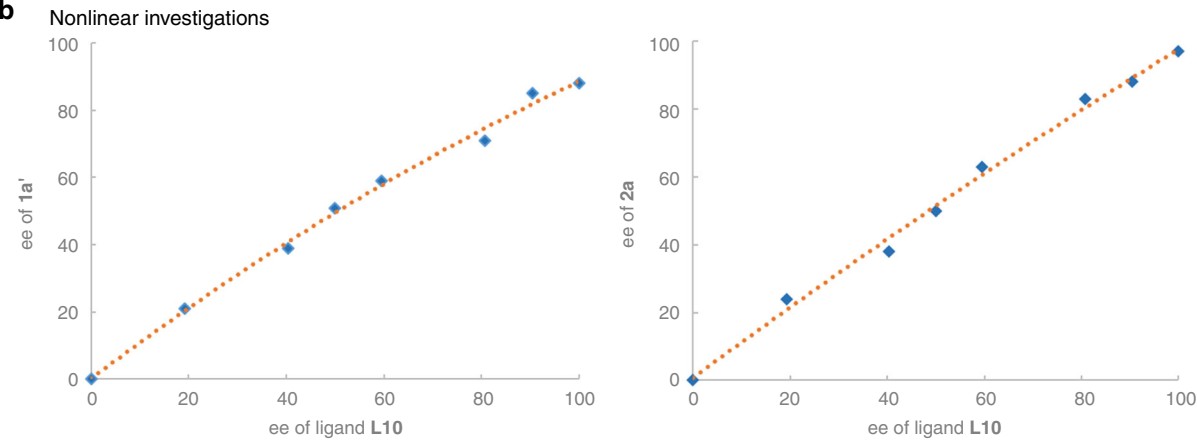

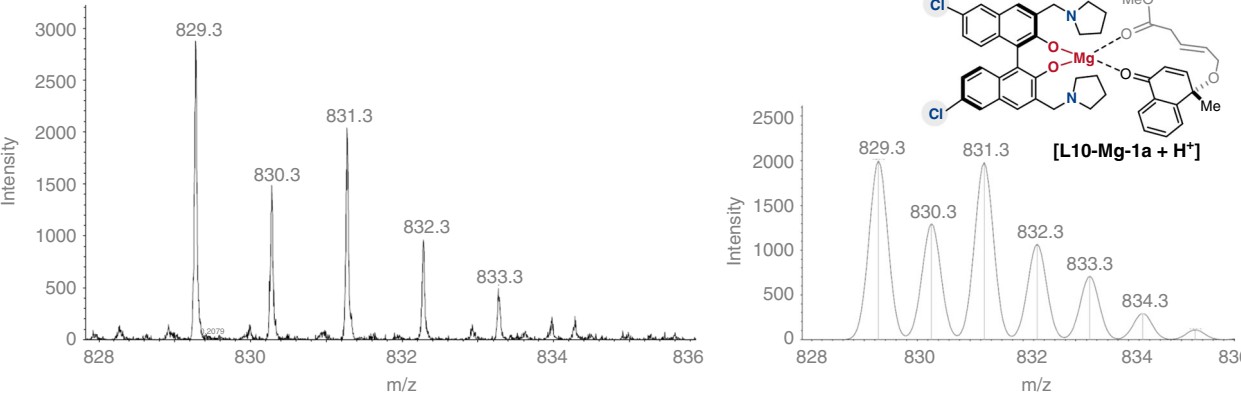

**Fig. 8 Mechanistic studies for the synergistic catalyst in the KR reaction. a** Control experiments of the KR reaction under different catalytic conditions. **b** Nonlinear effects studies of the KR reaction. **c** ESI experiments of the initial reaction complexes to investigate the activation mode.

proposal of a possible mechanism of this intramolecular vinylogous KR reaction. Further developments of the reported synergistic magnesium catalyst in asymmetric reactions are underway in our laboratory.

## Methods

**General procedure for the vinylogous KR reaction**. To a stirred solution of **L10** (10.42 mg, 0.02 mmol) in CPME (0.5 mL) was added Bu$_2$Mg (20 μL, 1.0 M in heptane, 0.02 mmol) under an argon atmosphere, the mixture was then stirred at room temperature for 30 min to generate the catalyst. The substrate **1** (0.2 mmol)

in CPME (0.5 mL) was quickly added to the flask containing the in situ generated magnesium catalyst. After the addition, the reaction was stirred at 40 °C and analyzed by TLC. The reaction was quenched with saturated NH$_4$Cl and extracted with CH$_2$Cl$_2$. The organic layer was dried over anhydrous Na$_2$SO$_4$ and concentrated under vacuum. Then the residue was purified by column chromatography to afford the product **1**[*] and **2**.

## Data availability

Detailed experimental procedures and characterization of compounds can be found in the Supplementary Information. The X-ray crystallographic coordinates for structures reported in this article have been deposited at the Cambridge Crystallographic Data

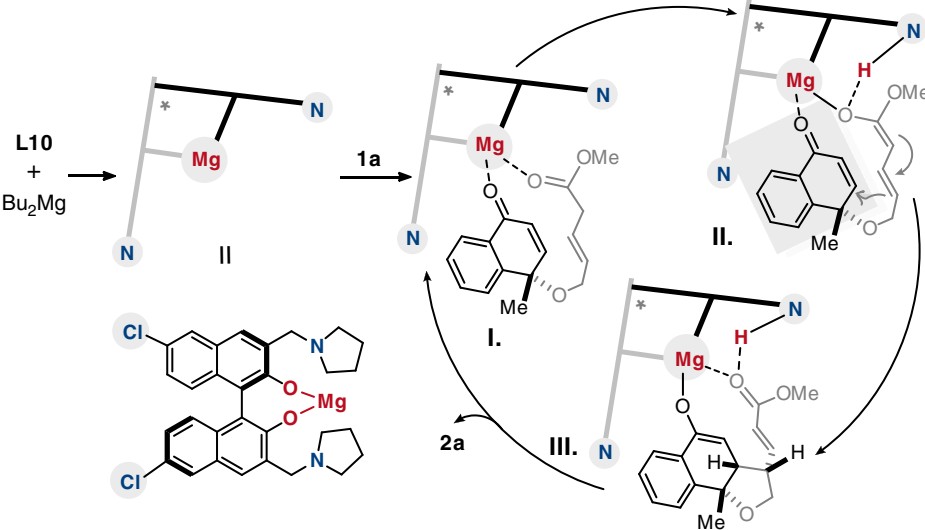

**Fig. 9 Proposed mechanism.** Possible reaction mechanism for the synergistic magnesium catalyst promoted KR reaction.

Center (**L10**: CCDC 1978958; **2b**: CCDC 1978955). These data could be obtained free of charge from The Cambridge Crystallographic Data Center via www.ccdc.cam.ac.uk/data_request/cif. All data are available from the authors upon request.

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

## Acknowledgements
We acknowledge the financial support from the NSFC (81473095, 21602091, 21901092), and the Fundamental Research Funds for the Central Universities (lzujbky-2019-68, 2018-kb11, 2017-19 and 2017-118). We would like to dedicate this to Professor Albert S. C. Chan on the occasion of his 70th birthday.

## Author contributions
D.L. carried out the major works on initial optimization, substrates and chiral ligands synthesis, substrates scope extension processes, mechanism study experiments and transformations of the products; D.L. analyzed the experiments data and prepared the Supplementary Information; Y.Y. extended parts of substrates and synthesized parts of substrates. M.Z. synthesized parts of the chiral ligands; L.W. synthesized parts of chiral ligands and analyzed the experiments data and the Supplementary Information; Y.X. carried out parts of HPLC and NMR works; D.Y. designed the reaction, wrote the manuscript and finished the arrangement works of the Supplementary Information; D.Y. and R.W. supervised the whole projects.

## Competing interests
The authors declare no competing interests.
