## [Peer Review File · Nature Communications]

Reviewers' comments:

Reviewer #1 (Remarks to the Author):

In this manuscript, Yang, Wang and coworkers report a magnesium-catalyzed kinetic resolution based on a designed intramolecular vinylogous Michael reaction. The reaction employs racemic naphthalen-1-ones bearing an allylic linear ester side chain as the substrates. With a newly synthesized BINOL-derived ligand containing two pyrrolidine groups, the corresponding Mg complex acts as a bifunctional catalyst. In addition to the activation of enone by Lewis acidic Mg center, the pyrrolidine moiety acts as a base to promote the deprotonation process. A series of experiments have been carried out to support the proposed mechanism. Under the optimized reaction conditions, a wide range of substrates could be well tolerated. Both the products and substrates are obtained in good yields (with 50% as maximal yield) and enantioselectivities. Particularly the site selective reaction of substrate 1s is truly interesting as the reaction seems to proceed at the sterically more hindered site. Some explanations here will be very useful. Indeed, the current work represents the first example of direct asymmetric vinylogous Michael reaction of linear allylic esters, and direct intramolecular asymmetric vinylogous reaction. These results certainly will inspire more research in this direction.

In addition, the gram-scale reaction and diverse transformations of the products are also demonstrated in the manuscript. These results clearly further showcase the synthetic potentials of this method in organic synthesis.

The SI is of high quality.

Overall, this is an excellent paper, and this reviewer would like to recommend the publication of this manuscript in *Nat. Commun.* after the following comments have been addressed.

Minor issues:

Page 3, line 3 from bottom, "catalysts," to "catalysts."

Since the current report is a kinetic resolution reaction, the authors might need to report the *S* value of the reaction. *S* value would provide straightforward view of the kinetic resolution process. When mentioning the asymmetric dearomatization reactions, a review on this topic should be added: *Angew. Chem. Int. Ed.* 2012, 51, 12662-12686.

Reviewer #2 (Remarks to the Author):

The manuscript by Yang, Wang and co-Workers describes asymmetric variant of intramolecular vinylogous Michael reaction through kinetic resolution by using synergistic magnesium catalyst. This work is interesting for the reported transformation as well as for the type of compounds that were synthesized. Although the asymmetric variant of the intramolecular vinylogous Michael reaction is rare, non-asymmetric intramolecular vinylogous Michael reaction has been reported: perhaps the authors can cite some of the recently reported papers on the intramolecular vinylogous conjugate additions including *Org. Lett.* 2019, 21, 1823; *J. Org. Chem.* 2010, 75, 3766; *Angew. Chem. Int. Ed.* 2006, 45, 2419.

They have designed substrates based on linear allylic esters suitable for asymmetric intramolecular vinylogous conjugate addition proceeding through kinetic resolution. This resulted in the synthesis of chiral [6.6.5]-tricyclic scaffolds in moderate yields with good enantioselectivities besides the remains of enantio-enriched starting allylic esters. The study is well done using a variety of substrates. All-carbon quaternary stereocenter was formed during the kinetic resolution of a representative substrate (1t) bearing two conjugate acceptor sites that underwent interesting site-selective Michael reaction. Gram-scale synthesis of 2a, enantio-enriched 1a and various transformations thereof were demonstrated that proved the practicality and synthetic utilities. The experimental work was conducted with care, with good characterization data and reasonable control experiments. Overall, the work is novel and is of good interest to the synthetic and

medicinal chemists. Based on these comments, this reviewer recommends this work for publication in Nature Communication after the minor revisions noted below.

The manuscript reads reasonably well, although it contains some minor errors and revisions that should be corrected/addressed before publication.

Page 3: "to the magnesium catalysts, The desired intramolecular vinylogous Michael reaction proceeded primary from one enantiomer, and resulted in enantiomerically enrich....." can be changed to "to the magnesium catalysts. The desired intramolecular vinylogous Michael reaction proceeded primarily from one enantiomer, and resulted in enantiomerically enrich....."

Page 5: "furnished the desired KR resolution" can be changed to "furnished the desired KR process"

Page 5: the ranges for the minor isomers may also be provided: "(92:8-98.5:1.5 er)"

Page 5: "Electro-withdrawing or donating groups" can be changed to "Electron-withdrawing or electron-donating groups"

Page 5: The authors reported that "The absolute configuration of the resolution adducts was determined by the X-ray crystallographic analysis of 2b". However, the assignment of the absolute configuration was not provided in the text or at the related structure in the Figure.

Importantly, the determination of the absolute configuration of 2b with X-ray arises some ambiguity, if the source of X-rays were MoK α radiations, as the compound 2b has no heavy atom in it. The authors may provide the source of X-rays (MoK α or CuK α radiations?) in the SI. The authors also should provide the Flack parameter in the supporting information. Please refer: "The use of X-ray Crystallography to Determine Absolute Configuration (II)" by Howard. D. Flack (*Acta Chim. Slov.* 2008, 55, 689). In case the authors used MoK α radiations, they should consider to provide X-ray structures of 1n* and 2n* that contain heavy atoms to avoid the ambiguity. Supporting Information: IR stretching frequencies can be rounded up to whole integers. In case of solid products, whether the IR was done as a KBr pellet, this information should be provided in the supporting information. Also, for most of the compounds IR values of 3300-3500 cm⁻¹ were reported, since there is no corresponding functional group present to indicate these values in the compounds -- the authors should verify this.

Supporting Information: for fluorine compounds, eg. in the cases of compound 1s, 1s*, 2s; multiplicity and coupling constants need to be provided in their respective ¹³NMR data.

Reviewer #3 (Remarks to the Author):

The manuscript details a kinetic resolution based on an intramolecular vinylogous Michael reaction. The process is catalysed by a magnesium catalyst coordinated by a bifunctional chiral BINOL-derived ligand, which possesses an additional Brønsted base function (a tertiary amine). The kinetic resolution process affords chiral [6.6.5]-tricyclic chiral skeletons with good selectivity. The chemistry has been designed assembling known reactivity concepts. Recently, Yin reported that a chiral copper catalyst, in the presence of bases, can promote the enolization of allylic esters upon coordination and deprotonation. This strategy has been then used to develop an asymmetric catalytic direct vinylogous aldol process (see Ref 40 in the manuscript – for a similar vinylogous activation of deconjugated carbonyl compounds, see *Angew. Chem. Int. Ed.* 2019, 58, 9210–9214). Here a similar synergistic approach has been used, where a chiral Lewis acid (magnesium instead of copper) promotes the enolization of similar ester substrates. The difference is that, in the present study, the Brønsted base that assists the enolization is placed within the chiral ligand scaffold. This approach is however not without precedent, for the authors have already demonstrated that a chiral magnesium catalyst, where the basic site was on the chiral ligand (*Angew. Chem. Int. Ed.* 2013, 52, 6739 –6742), could effectively drive the enolization of enone substrates, which then reacted through a vinylogous pathway.

The only element of novelty here is that the vinylogous chemistry of the dienolates, generated upon deprotonation of allylic esters, is used in an intramolecular Michael addition. Alone, this aspect is interesting but does not justify publication in a general chemistry journal, in particular

when considering that intramolecular kinetic resolution processes have been already reported (Refs 5-14).

From a synthetic standpoint, the reaction requires the preparation of tailored substrates, thus lowering the generality of the approach.

Overall, this manuscript lacks the marks of originality and general synthetic interest required for publication on a general chemistry journal. This study seems well suited for a more specialised, synthetically-oriented audience.

Other comment: the stereoselectivity of a kinetic resolution process is assessed by measuring the selectivity factor. Since the enantiomeric excess of the products and the unreacted substrates changes with the conversion, this is the only meaningful way to describe the stereoselectivity. The selectivity factor should be accurately calculated and discussed.

Reviewers' comments and corresponding responses:

Reviewers' comments:

Reviewer #1 (Remarks to the Author):

In this manuscript, Yang, Wang and coworkers report a magnesium-catalyzed kinetic resolution based on a designed intramolecular vinylogous Michael reaction. The reaction employs racemic naphthalen-1-ones bearing an allylic linear ester side chain as the substrates. With a newly synthesized BINOL-derived ligand containing two pyrrolidine groups, the corresponding Mg complex acts as a bifunctional catalyst. In addition to the activation of enone by Lewis acidic Mg center, the pyrrolidine moiety acts as a base to promote the deprotonation process. A series of experiments have been carried out to support the proposed mechanism. Under the optimized reaction conditions, a wide range of substrates could be well tolerated. Both the products and substrates are obtained in good yields (with 50% as maximal yield) and enantioselectivities. Particularly the site selective reaction of substrate 1s is truly interesting as the reaction seems to proceed at the sterically more hindered site. Some explanations here will be very useful.

Indeed, the current work represents the first example of direct asymmetric vinylogous Michael reaction of linear allylic esters, and direct intramolecular asymmetric vinylogous reaction. These results certainly will inspire more research in this direction.

In addition, the gram-scale reaction and diverse transformations of the products are also demonstrated in the manuscript. These results clearly further showcase the synthetic potentials of this method in organic synthesis.

The SI is of high quality.

Overall, this is an excellent paper, and this reviewer would like to recommend the publication of this manuscript in Nat. Commun. after the following comments have been addressed.

Minor issues:

Page 3, line 3 from bottom, "catalysts," to "catalysts."

Since the current report is a kinetic resolution reaction, the authors might need to report the S value of the reaction. S value would provide straightforward view of the kinetic resolution process.

When mentioning the asymmetric dearomatization reactions, a review on this topic should be added: Angew. Chem. Int. Ed. 2012, 51, 12662-12686.

Reviewer #2 (Remarks to the Author):

The manuscript by Yang, Wang and co-Workers describes asymmetric variant of intramolecular vinylogous Michael reaction through kinetic resolution by using synergistic magnesium catalyst. This work is interesting for the reported transformation as well as for the type of compounds that were synthesized. Although the asymmetric variant of the intramolecular vinylogous Michael reaction is rare, non-asymmetric intramolecular vinylogous Michael reaction has been reported: perhaps the authors can cite some of the recently reported papers on the intramolecular vinylogous conjugate additions including *Org. Lett.* 2019, 21, 1823; *J. Org. Chem.* 2010, 75, 3766; *Angew. Chem. Int. Ed.* 2006, 45, 2419.

They have designed substrates based on linear allylic esters suitable for asymmetric intramolecular vinylogous conjugate addition proceeding through kinetic resolution. This resulted in the synthesis of chiral [6.6.5]-tricyclic scaffolds in moderate yields with good enantioselectivities besides the remains of enantio-enriched starting allylic esters. The study is well done using a variety of substrates. All-carbon quaternary stereocenter was formed during the kinetic resolution of a representative substrate (1t) bearing two conjugate acceptor sites that underwent interesting site-selective Michael reaction. Gram-scale synthesis of 2a, enantio-enriched 1a and various transformations thereof were demonstrated that proved the practicality and synthetic utilities. The experimental work was conducted with care, with good characterization data and reasonable control experiments. Overall, the work is novel and is of good interest to the synthetic and medicinal chemists. Based on these comments, this reviewer recommends this work for publication in *Nature Communication* after the minor revisions noted below.

The manuscript reads reasonably well, although it contains some minor errors and revisions that should be corrected/addressed before publication.

Page 3: "to the magnesium catalysts, The desired intramolecular vinylogous Michael reaction proceeded primary from one enantiomer, and resulted in enantiomerically enrich....." can be changed to "to the magnesium catalysts. The desired intramolecular vinylogous Michael reaction proceeded primarily from one enantiomer, and resulted in enantiomerically enrich....."

Page 5: "furnished the desired KR resolution" can be changed to "furnished the desired KR process"

Page 5: the ranges for the minor isomers may also be provided: "(92:8-98.5:1.5 er)"

Page 5: "Electro-withdrawing or donating groups" can be changed to "Electron-withdrawing or electron-donating groups"

Page 5: The authors reported that "The absolute configuration of the resolution adducts was determined by the X-ray crystallographic analysis of 2b". However, the assignment of the absolute configuration was not provided in

the text or at the related structure in the Figure.

Importantly, the determination of the absolute configuration of 2b with X-ray arises some ambiguity, if the source of X-rays were MoK α radiations, as the compound 2b has no heavy atom in it. The authors may provide the source of X-rays (MoK α or CuK α radiations?) in the SI. The authors also should provide the Flack parameter in the supporting information. Please refer: "The use of X-ray Crystallography to Determine Absolute Configuration (II)" by Howard. D. Flack (Acta Chim. Slov. 2008, 55, 689). In case the authors used MoK α radiations, they should consider to provide X-ray structures of 1n* and 2n* that contain heavy atoms to avoid the ambiguity.

Supporting Information: IR stretching frequencies can be rounded up to whole integers. In case of solid products, whether the IR was done as a KBr pellet, this information should be provided in the supporting information. Also, for most of the compounds IR values of 3300-3500 cm⁻¹ were reported, since there is no corresponding functional group present to indicate these values in the compounds -- the authors should verify this.

Supporting Information: for fluorine compounds, eg. in the cases of compound 1s, 1s*, 2s; multiplicity and coupling constants need to be provided in their respective 13-NMR data.

Reviewer #3 (Remarks to the Author):

The manuscript details a kinetic resolution based on an intramolecular vinylogous Michael reaction. The process is catalysed by a magnesium catalyst coordinated by a bifunctional chiral BINOL-derived ligand, which possesses an additional Brønsted base function (a tertiary amine). The kinetic resolution process affords chiral [6.6.5]-tricyclic chiral skeletons with good selectivity.

The chemistry has been designed assembling known reactivity concepts. Recently, Yin reported that a chiral copper catalyst, in the presence of bases, can promote the enolization of allylic esters upon coordination and deprotonation. This strategy has been then used to develop an asymmetric catalytic direct vinylogous aldol process (see Ref 40 in the manuscript – for a similar vinylogous activation of deconjugated carbonyl compounds, see Angew. Chem. Int. Ed. 2019, 58, 9210–9214). Here a similar synergistic approach has been used, where a chiral Lewis acid (magnesium instead of copper) promotes the enolization of similar ester substrates. The difference is that, in the present study, the Brønsted base that assists the enolization is placed within the chiral ligand scaffold. This approach is however not without precedent, for the authors have already demonstrated that a chiral magnesium catalyst, where the basic site was on the chiral ligand (Angew. Chem. Int. Ed. 2013, 52, 6739–6742), could effectively drive the enolization of enone substrates, which then reacted through a vinylogous pathway.

The only element of novelty here is that the vinylogous chemistry of the dienolates, generated upon deprotonation of allylic esters, is used in an intramolecular Michael addition. Alone, this aspect is interesting but does not justify publication in a general chemistry journal, in particular when considering that intramolecular kinetic resolution processes have been already reported (Refs 5-14).

From a synthetic standpoint, the reaction requires the preparation of tailored substrates, thus lowering the generality of the approach.

Overall, this manuscript lacks the marks of originality and general synthetic interest required for publication on a general chemistry journal. This study seems well suited for a more specialised, synthetically-oriented audience.

Other comment: the stereoselectivity of a kinetic resolution process is assessed by measuring the selectivity factor. Since the enantiomeric excess of the products and the unreacted substrates changes with the conversion, this is the only meaningful way to describe the stereoselectivity. The selectivity factor should be accurately calculated and discussed.

Corresponding responses to the reviewers:

Responses to reviewer #1:

Suggestion 1: *Particularly the site selective reaction of substrate 1s is truly interesting as the reaction seems to proceed at the sterically more hindered site. Some explanations here will be very useful.*

Response 1: As the reviewer mentioned, the product isolated from this substrate is generated at the sterically more hindered site, and we noted some particular transformations can occur at the more hindered site under specific catalytic conditions in dienone structures as the literature we cited in ref 56 in the revised manuscript. But there are also some unidentified products are obtained for this substrate's reaction, which are not clean enough, so we speculated the cyclization product at the expected site might not be stable enough to be separated, but we are not very clear to the structure of the unidentified decomposition products for there might be also other reaction pathways occur such as dienone rearrangement for the particular structure of this substrate. So we have not given confirmed explanation in the manuscript and just claimed "The common cyclization adduct **2t'** was not observed under the catalytic system, instead, some undetermined decomposition products were generated, resulting in the relatively lower yields of **2t** and **1t***" in the manuscript. Thanks for the helpful suggestions of the reviewer.

Suggestion 2: *Indeed, the current work represents the first example of direct asymmetric vinylogous Michael reaction of linear allylic esters, and direct intramolecular asymmetric vinylogous reaction. These results certainly will inspire more research in this direction.*

In addition, the gram-scale reaction and diverse transformations of the products are also demonstrated in the manuscript. These results clearly further showcase the synthetic potentials of this method in organic synthesis.

The SI is of high quality.

Overall, this is an excellent paper, and this reviewer would like to recommend the publication of this manuscript in Nat. Commun. after the following comments have been addressed.

Minor issues:

Page 3, line 3 from bottom, "catalysts," to "catalysts."

Since the current report is a kinetic resolution reaction, the authors might need to report the S value of the reaction. S value would provide straightforward view of the kinetic resolution process.

When mentioning the asymmetric dearomatization reactions, a review on this topic should be added: Angew. Chem. Int. Ed. 2012, 51, 12662-12686.

Response 2: Thanks for the reviewer's positive comments. The mistake on page 3 has been revised. And the s values are provided except for the particular reaction of the substrate in Figure 5 in the revised manuscript, for there are major unidentified byproducts generated, which is also claimed, and the related definition of the s value is provided in the footnote c of Table 1. The important review on asymmetric dearomatization reactions is cited in ref 20 in the revised manuscript. Thanks for all of these helpful suggestions by the reviewer.

Responses to reviewer #2:

Suggestion 1: *The manuscript by Yang, Wang and co-Workers describes asymmetric variant of intramolecular vinylogous Michael reaction through kinetic resolution by using synergistic magnesium catalyst. This work is interesting for the reported transformation as well as for the type of compounds that were synthesized. Although the asymmetric variant of the intramolecular vinylogous Michael reaction is rare, non-asymmetric intramolecular vinylogous Michael reaction has been reported: perhaps the authors can cite some of the recently reported papers on the intramolecular vinylogous conjugate additions including Org. Lett. 2019, 21, 1823; J. Org. Chem. 2010, 75, 3766; Angew. Chem. Int. Ed. 2006, 45, 2419.*

Response 1: Thanks for the reviewer's positive comments, and the corresponding papers on racemic intramolecular Michael reactions are cited in the revised manuscript at refs 42-44. Thanks for the reviewer's detailed suggestions.

Suggestion 2: *The manuscript reads reasonably well, although it contains some minor errors and revisions that should be corrected/addressed before publication.*

Page 3: "to the magnesium catalysts, The desired intramolecular vinylogous Michael reaction proceeded primary from one enantiomer, and resulted in enantiomerically enrich....." can be changed to "to the magnesium catalysts. The desired intramolecular vinylogous Michael reaction proceeded primarily from one enantiomer, and resulted in enantiomerically enrich....."

Page 5: "furnished the desired KR resolution" can be changed to "furnished the desired KR process"

Page 5: the ranges for the minor isomers may also be provided: "(92:8-98.5:1.5 er)"

Page 5: "Electro-withdrawing or donating groups" can be changed to "Electron-withdrawing or electron-donating groups"

Page 5: The authors reported that "The absolute configuration of the resolution adducts was determined by the X-ray crystallographic analysis of 2b". However, the assignment of the absolute configuration was not provided in the text or at the related structure in the Figure.

Response 2: The mistakes pointed out by the reviewer are revised. And in the transformations we have not identified the minor isomer both for the Z-isomer or the diastereoisomers of products 2, which may be attributing to the transition state of the intramolecular reaction pathway and activation mode of the substrates with the bifunctional magnesium catalyst. The absolute configuration of the product 2 is determined by the X-ray crystallographic analysis of compound 2b, the related X-ray structure of 2b is provided in the Figure 3 besides the product structures of 2a-2d, and a bracket note of (Fig. 3) is also provided after the corresponding sentence. Thanks for the reviewer's detail suggestions and we are grateful for these advices.

Suggestion 3: *Importantly, the determination of the absolute configuration of 2b with X-ray arises some ambiguity, if the source of X-rays were MoK α radiations, as the compound 2b has no heavy atom in it. The authors may provide the source of X-rays (MoK α or CuK α radiations?) in the SI. The authors also should provide the Flack parameter in the supporting information. Please refer: "The use of X-ray Crystallography to Determine Absolute Configuration (II)" by Howard. D. Flack (Acta Chim. Slov. 2008, 55, 689). In case the authors used MoK α radiations,*

they should consider to provide X-ray structures of 1n and 2n* that contain heavy atoms to avoid the ambiguity.*

Response 3: The absolute configuration of the products is determined by the X-ray crystallographic analysis of product 2b. "The X-Ray diffraction of Cu K α radiation was used, the absolute configuration was unequivocally determined through anomalous dispersion effects with a Flack x parameter of 0.02(11). According to Howard. D. Flack, The use of X-ray Crystallography to Determine Absolute Configuration (II). Acta Chim. Slov. 55, 689–691 (2008)." These information are included in the revised supporting information on page 24, and the corresponding Flack parameter of ligand L10 is also provided on page 37 in the revised supporting information. Thanks for the reviewer's important advice.

Suggestion 4: *Supporting Information: IR stretching frequencies can be rounded up to whole integers. In case of solid products, whether the IR was done as a KBr pellet, this information should be provided in the supporting information. Also, for most of the compounds IR values of 3300-3500 cm⁻¹ were reported, since there is no corresponding functional group present to indicate these values in the compounds -- the authors should verify this. Supporting Information: for fluorine compounds, eg. in the cases of compound 1s, 1s*, 2s; multiplicity and coupling constants need to be provided in their respective 13-NMR data.*

Response 4: The IR data are rounded up to whole integers. And the IR was done by using KBr pallet, these information are included in the supporting information. And the peaks between 3300-3500 cm⁻¹ are found in the IR spectra as it is showed with the IR spectra of compounds 1a* and 2a provided in the Figure 1 in the response letter, we also can not confirm it to the exact functional groups, but we have found for some unsaturated or allylic carbonyl compounds in reported literatures, the peaks between 3300-3500 cm⁻¹ can also be found, such as a case reported recently: Wang et al. Catalytic enantioselective oxidative coupling of saturated ethers with carboxylic acid derivatives. Nat Commun. 2019, 10, 559. And the fluorine compounds, including compound 1s, 1s*, 2s; multiplicity and coupling constants are provided in their respective 13-NMR data. Thanks for the reviewer's detail and important suggestions.

Figure 1 in the response letter. IR spectra of 1a (left) and 2a (right).

Responses to reviewer #3:

Suggestion 1: *The chemistry has been designed assembling known reactivity concepts. Recently, Yin reported that a chiral copper catalyst, in the presence of bases, can promote the enolization of allylic esters upon coordination and deprotonation. This strategy has been then used to develop an asymmetric catalytic direct vinylogous aldol process (see Ref 40 in the manuscript – for a similar vinylogous activation of deconjugated carbonyl compounds, see *Angew. Chem. Int. Ed.* 2019, 58, 9210–9214). Here a similar synergistic approach has been used, where a chiral Lewis acid (magnesium instead of copper) promotes the enolization of similar ester substrates. The difference is that, in the present study, the Brønsted base that assists the enolization is placed within the chiral ligand scaffold. This approach is however not without precedent, for the authors have already demonstrated that a chiral magnesium catalyst, where the basic site was on the chiral ligand (*Angew. Chem. Int. Ed.* 2013, 52, 6739–6742), could effectively drive the enolization of enone substrates, which then reacted through a vinylogous pathway.*

Response 1: Thanks for the reviewer's comments on this work. The works mentioned by the reviewer were well cited and illustrated in the manuscript, for the citation of Yin's work can be found in the introduction part and related references (corresponding ref 37-40 in the former submitted manuscript and ref 38-41 in this revised one), the introduction part clearly illustrate the differences and novelties of our work as the first direct intramolecular asymmetric vinylogous Michael reaction of linear allylic esters, and the catalyst we used is totally different with Yin's published works (Phosphine ligands were used in Yin's works), and the substrates in our work are new

designed compounds. The corresponding novelties of our work are also commented by reviewer 1 and 2.

Moreover, the reviewer mentioned the work is similar to us: *Angew. Chem. Int. Ed.* 2019, 58, 9210-9214, which has also been cited in our former cited manuscript in the introduction part (ref 28 in our former submitted manuscript and ref 29 in the current revised version), this work is a totally different reaction pathway and used different catalytic methods as shown in the following figure (Figure 2 in the response letter). The reference reaction is a Pd-mediated asymmetric allylic alkylation (AAA) reaction, most allylic carbonyl compounds are ketones and only one allylic ester case is reported, more importantly, the catalytic methods, the mechanism and the reactive site of the reaction partner is all different as shown in the followed Figure.

Figure 2 in the response letter: difference between our work and the reference work.

For the reference: *Angew. Chem. Int. Ed.* 2013, 52, 6739-6742, which is also cited in the introduction part (ref 46 in our former submitted manuscript and ref 50 in the current revised version), the reaction is our former work on intermolecular [4+2] cyclization reaction between of enones and nitroalkenes, the current is an intramolecular vinylogous Michael reaction of linear allylic esters, and the chiral ligand used in current work is totally different, the ligand in this reference used Salen ligand, and here we developed new Binolated derived bifunctional chiral ligands, the synthesis methods of the current chiral ligand is also provided in the manuscript. Thanks for the detail comments of the reviewer.

Suggestion 2: *The only element of novelty here is that the vinylogous chemistry of the dienolates, generated upon deprotonation of allylic esters, is used in an intramolecular Michael addition. Alone, this aspect is interesting but does not justify publication in a general chemistry journal, in particular when considering that intramolecular kinetic resolution processes have been already reported (Refs 5-14).*

Response 2: The current work is the catalytic asymmetric intramolecular vinylogous Michael reaction, which is also commented by the reviewer 1 and reviewer 2. And the refs 5-14 are intramolecular kinetic resolution reactions we have cited, the reaction pathways and related mechanism in these references are almost totally different from current work. Thanks for the reviewer's related comments.

Suggestion 3: *From a synthetic standpoint, the reaction requires the preparation of tailored substrates, thus lowering the generality of the approach.*

Overall, this manuscript lacks the marks of originality and general synthetic interest required for publication on a general chemistry journal. This study seems well suited for a more specialised, synthetically-oriented audience.

Response 3: The substrates are new designed compounds, the designed reaction of these substrates are used to synthesis the key chiral tricyclic skeletons under the developed magnesium catalyst, as it is also commented by reviewer 2. The reaction is the first direct asymmetric intramolecular vinylogous Michael reaction of linear allylic esters, which can inspire other vinylogous conjugate reaction as well, as it is also commented by the reviewer 1, and we have also developed other asymmetric vinylogous reactions on the basis of this work.

And other novelties of this manuscript is illustrated in the manuscript and the related responses to the suggestions 1 and 2 of this reviewer as aforementioned, as well as in the comments from the reviewer 1 and reviewer 2. Thanks to the reviewer's related comments.

Suggestion 4: *Other comment: the stereoselectivity of a kinetic resolution process is assessed by measuring the selectivity factor. Since the enantiomeric excess of the products and the unreacted substrates changes with the conversion, this is the only meaningful way to describe the stereoselectivity. The selectivity factor should be accurately calculated and discussed.*

Response 4: The selectivity factor is added in the revised manuscript for the kinetic resolution reaction except the substrate 1t, for there are unidentified other decomposition products in the reaction of this special substrate, as we claimed in the manuscript, and the related definition of the s value is provided in the footnote c of Table 1. Thanks for the detail suggestions by the reviewer.

REVIEWERS' COMMENTS:

Reviewer #1 (Remarks to the Author):

In this revised manuscript, Yang, Wang and coworkers did a great job to address the comments from this reviewer. This reviewer would like to recommend the acceptance of the revised manuscript in Nature Comm.

A minor point: Just in case I miss it, the information on how to calculate the S value should be added in the MS or SI.

Reviewer #2 (Remarks to the Author):

This revised manuscript submitted by by Yang, Wang and co-Workers describes "Activation of Allylic Esters in an Intramolecular Vinylogous Kinetic Resolution Reaction with Synergistic Magnesium Catalysts".

The authors have addressed the major concerns/corrections/suggestions raised by the reviewers. I would like to recommend the publication of this work in Nature Communications journal. However, below I have noted some of typos/errors/suggestion for the attention of the Authors before publication.

Line 66: 'baring' to 'bearing'

Line 86: 'perform' to 'performed'

Line 107: 'Discussion' or 'Conclusion'?

Line 108: 'direct catalytic intramolecular vinylogous Michael' to 'direct catalytic asymmetric intramolecular vinylogous Michael'

Line 110: 'successfully lead' to 'successfully led'

Line 123: 'dried over Na₂SO₄' to 'dried over anhydrous Na₂SO₄'

The IR stretching frequencies in the 3300-3500 cm⁻¹: are these due to the moisture present in KBr or the products? As any carbonyl stretch may not be observed during this region, the Authors may please cross-check this once again.

Corresponding responses to the reviewers:

Responses to reviewer #1:

Suggestion: In this revised manuscript, Yang, Wang and coworkers did a great job to address the comments from this reviewer. This reviewer would like to recommend the acceptance of the revised manuscript in Nature Comm. A minor point: Just in case I miss it, the information on how to calculate the S value should be added in the MS or SI.

Response: Thanks for the kind attentions and related detailed suggestions. The calculated method of s value is added in the footnote part of Table 1 and also in the legend of Fig. 3 accordingly. Thanks again for the detailed and helpful suggestions.

Responses to reviewer #2:

Suggestion: This revised manuscript submitted by by Yang, Wang and co-Workers describes "Activation of Allylic Esters in an Intramolecular Vinylogous Kinetic Resolution Reaction with Synergistic Magnesium Catalysts".

The authors have addressed the major concerns/corrections/suggestions raised by the reviewers. I would like to recommend the publication of this work in Nature Communications journal.

However, below I have noted some of typos/errors/suggestion for the attention of the Authors before publication.

Line 66: 'baring' to 'bearing'

Line 86: 'perform' to 'performed'

Line 107: 'Discussion' or 'Conclusion'?

Line 108: 'direct catalytic intramolecular vinylogous Michael' to 'direct catalytic asymmetric intramolecular vinylogous Michael'

Line 110: 'successfully lead' to 'successfully led'

Line 123: 'dried over Na₂SO₄' to 'dried over anhydrous Na₂SO₄'

The IR stretching frequencies in the 3300-3500 cm⁻¹: are these due to the moisture present in KBr or the products?

As any carbonyl stretch may not be observed during this region, the Authors may please cross-check this once again.

Response: Thanks for the kind attentions and many helpful suggestions. The mistakes pointed out by the reviewer are revised. And we noted that the format of the MS often include a Discussion part, so the Discussion is used. And we have rechecked the IR experiments, the stretching frequencies in 3300-3500 cm⁻¹ for the

products should be according to moisture during the diluted process of samples before IR experiments. The related data in supporting information is also revised. Thanks for the reviewer's helpful and important suggestions on our work.